

# Consequences for selected high-elevation butterflies and moths from the spread of *Pinus mugo* into the alpine zone in the High Sudetes Mountains

Karolína Bílá[1,2], Jan Šipoš[3], Pavel Kindlmann[2,4] and Tomáš Kuras[1]

[1] Department of Ecology and Environmental Sciences, Faculty of Science, Palacký University Olomouc, Olomouc, Czech Republic
[2] Department of Biodiversity Research, Global Change Research Institute CAS, Brno, Czech Republic
[3] Department of Biology and Ecology, Institute of Environmental Technologies, Faculty of Science, University of Ostrava, Ostrava, Czech Republic
[4] Institute of Environmental Studies, Charles University Prague, Prague, Czech Republic

Corresponding author
Karolína Bílá, kcerna@volny.cz

## ABSTRACT

Due to changes in the global climate, isolated alpine sites have become one of the most vulnerable habitats worldwide. The indigenous fauna in these habitats is threatened by an invasive species, dwarf pine (*Pinus mugo*), which is highly competitive and could be important in determining the composition of the invertebrate community. In this study, the association of species richness and abundance of butterflies with the extent of *Pinus mugo* cover at individual alpine sites was determined. Butterflies at alpine sites in the High Sudetes Mountains (Mts.) were sampled using Moericke yellow water traps. The results of a Canonical Correspondence Analysis (CCA) indicated that at a local scale the area of alpine habitats is the main limiting factor for native species of alpine butterflies. Butterfly assemblages are associated with distance to the tree-line with the optimum situated in the lower forest zone. In addition the CCA revealed that biotic factors (i.e. *Pinus mugo* and alpine tundra vegetation) accounted for a significant amount of the variability in species data. Regionally, the CCA identified that the species composition of butterflies and moths is associated with presence and origin of *Pinus mugo*. Our study provides evidence that the structure of the Lepidopteran fauna that formed during the postglacial period and also the present composition of species assemblages is associated with the presence of *Pinus mugo*. With global warming, *Pinus mugo* has the potential to spread further into alpine areas and negatively affect the local species communities.

# INTRODUCTION

Alpine habitats are unique and highly vulnerable because their existence depends on abiotic conditions, island phenomena and postglacial history (*Taberlet et al.*, *1998*; *Schmitt & Haubrich*, *2008*). These habitats have been affected by changes in global climate and anthropogenic activities, which favour the spread of non-indigenous species (*Nagy & Grabherr*, *2009*). However, there are few studies on the effect of biotic factors on the

composition of the community in this habitat, which developed during the postglacial period. It is highly likely that competitive species determined the structure of the community and survival of other species (*Maron & Vilà*, *2001*).

Dwarf pine (*Pinus mugo*) is a typical species of Central European mountains where it is usually dominant at the lower margin of the subalpine altitudinal zone. Dwarf pine limits the distribution of many other plants and animals within the alpine zone (*Cavalli et al.*, *2011*; *Zeidler et al.*, *2012*). This successful colonist spreads not only at localities within its indigenous distribution area but particularly on mountain summits where it was artificially planted (*Dullinger, Dirnböck & Grabherr*, *2003*; *Treml et al.*, *2010*). The spread of dwarf pine is negatively affecting alpine zones by overgrowing the native treeless area, fragmenting and increasing its isolation, decreasing the amount of food available for insects and changing the microclimate (*Svoboda*, *2001*).

As far as we know, only one study records the negative effects of *Pinus mugo* on the indigenous fauna, in particular beetles (*Kašák et al.*, *2015*). Here, we present recent findings and hypothesize that *Pinus mugo* has a negative effect on Lepidoptera richness and species composition at both, local and regional levels. Both these effects of *Pinus mugo* are discussed in terms of recent and postglacial development of invertebrate communities.

At the local level, we estimate the effects of (i) habitat quality, (ii) area of the alpine site and (iii) edge effect on species richness of butterflies; at the regional level, we estimate the effects of (i) the extent of the area covered by *Pinus mugo* (naturally occurring and planted), (ii) *Pinus mugo* origin, and (iii) area of the alpine site, on both butterflies and moths.

## MATERIALS AND METHODS

### Ethics statement
Our field survey was approved by the Czech Ministry of Environment and we were allowed to collect species of Lepidoptera at all alpine sites studied, which are part of the Protected Landscape area Jeseníky. The permission was issued for a group of scientists, one of which was Dr. Tomáš Kuras. Permission Nr.: MŽP/13341/04-620/2319/04 from the 21st of July 2004, valid untill the 30th of November 2007.

### Study area
The High Sudetes Mts. form a natural border between the Czech Republic and Poland (*Jeník*, *1961*) and are one of the middle-high mountain systems in Central Europe. There are sites in the alpine zone in the High Sudetes Mts. with different degrees of isolation and different *Pinus mugo* origin; i.e., these mountains are at the edge of the distribution of *Pinus mugo* with some sites occupied by this indigenous dwarf pine and others where this shrub is absent or was artificially planted. In the Czech Republic, the High Sudetes Mts. consist of three mountain ranges (Fig. 1): the Krkonoše Mts. (Riesengebirge/Giant Mts.), Hrubý Jeseník Mts. (Altvatergebirge) and Kralický Sněžník Mts. (Glatzer Schneegebirge). *Pinus mugo* occurs naturally in the Krkonoše Mts., but not in the Kralický Sněžník and Hrubý Jeseník Mts.

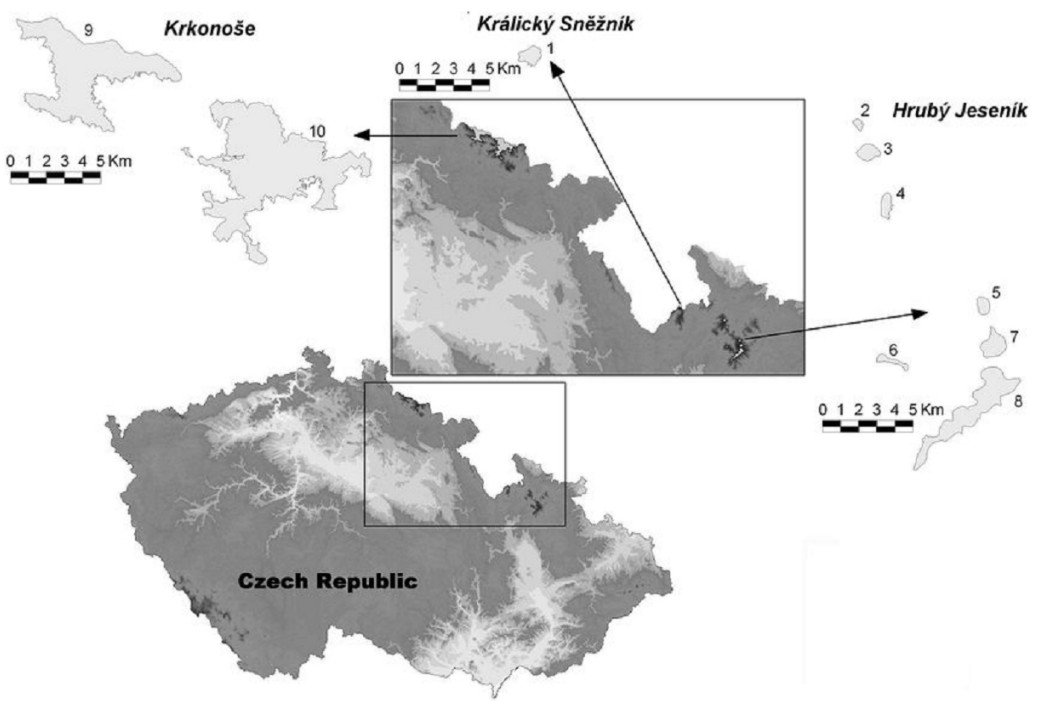

**Figure 1** Map of the High Sudetes Mts., which consist of three mountain ranges (Krkonoše Mts., Kralický Sněžník Mts. and Hrubý Jeseník Mts.), showing the location of the alpine sites studied (abbreviations used in Fig. 6 in brackets). 1, Kralický Sněžník (Kra.Sneznik); 2, Šerák (Serak); 3, Keprník (Keprnik); 4, Červená hora (Cer.hora); 5, Malý Děd (Mal.Ded); 6, Mravenečník-Vřesník (Mr.Vresnik); 7, Praděd (Praded); 8, Vysoká hole (Vys.hole); 9, Krkonoše-West (Krko.W); and 10, Krkonoše-East (Krko.E).

We also studied sites at a wider regional scale by including the two geographically nearest alpine habitats of different origins, with *Pinus mungo* present or absent; i.e., Babia Gora, Slovakia/Poland with *Pinus mugo* and Harz, Germany, without *Pinus mugo* (Fig. 2).

## Historical context

Summits in the High Sudetes usually reach an altitude of not more than 1,600 m and have a narrow belt of treeless arctic-alpine tundra at the summits. Although the insect fauna of middle-high mountains is typically poor compared to that at lower altitudes (*Dennis, Shreeve & Williams*, 1995; *Fleishman, Austin & Weiss*, 1998; *Nagy et al.*, 2003), alpine sites in the High Sudetes Mts. host a unique fauna comprised of arcto-alpine and boreo-montane species, plus endemic species and glacial relicts (*Mazalová, Kašák & Kuras*, 2012). A similar fauna does not occur anywhere else in Central Europe and the sites studied in the High Sudetes Mts. strongly differ in species composition. Therefore, the High Sudetes have a high conservation value. The exceptional nature of these mountains is associated with their location near the northern-most border of *Pinus mugo* distribution (*Hamerník & Musil*, 2007). Dwarf pine is indigenous in most Central European mountains (incl. Western Carpathians, Beskydy Mts., Krkonoše Mts./Riesengebirge) and artificially planted in the Hrubý Jeseník Mts./Altvatergebirge and the Kralický Sněžník Mts./Glatzer Schneegebirge (*Klimeš & Klimešová*, 1991; *Rybníček & Rybníčková*, 2004). Afforestation of most of the

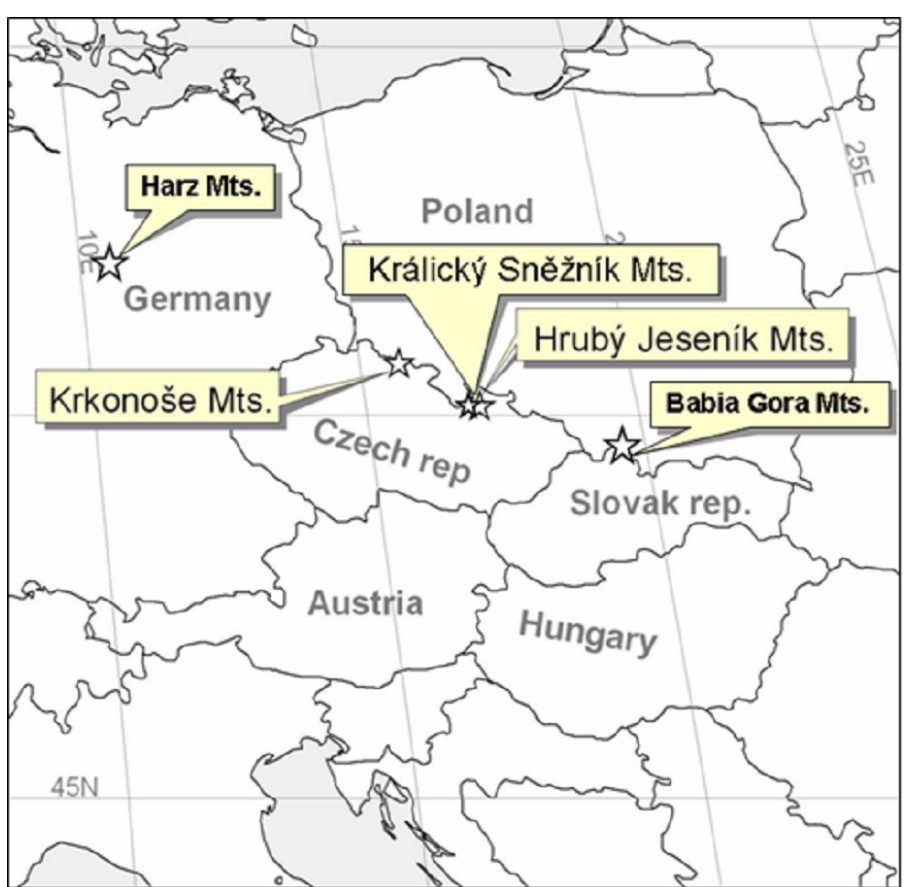

**Figure 2** Map showing the areas studied at the Central European scale.

summits in Hrubý Jeseník Mts. and Kralický Sněžník with *Pinus mugo* started at the end of the 19th century (*Jeník & Hampel*, *1992*) and resulted in it colonizing the area where it currently covers about 63% of what was previously alpine habitat (*Treml et al.*, *2010*). On the other hand, this latter afforestation allows us to test recent vs. postglacial effects of *Pinus mugo*.

## Insect groups studied and sampling method

A survey of butterflies was carried out in the Hrubý Jeseník and the Kralický Sněžník Mts.; a group of invertebrates with high conservation and bio-indication importance. Butterflies were captured using yellow plastic traps ("Moericke water traps"), which are widely used for monitoring nectar feeding insects and thus also suitable for butterflies (*Barták*, *1997*; *Roháček, Barták & Kubík*, *1998*; *Beneš, Kuras & Konvička*, *2000*). The traps were made of circular plastic pans (upper diameter: 13 cm, depth: 6 cm) painted yellow inside (Industrol 6200®). At the study sites the pans were filled to a depth of about 2 cm with a solution of detergent and water (1 ml of detergent per 1 l of water). We placed the traps approximately 20 m apart along transects at each study site, with the number of traps set dependent on the area of the site. The traps were set during the flight period of the butterflies (*Kuras et al.*, *2001b*, *Kuras et al.*, *2003*; *Konvička, Beneš & Kuras*, *2002*), i.e., from July 15 to September 3
**Table 1** Trapping sites, number of traps and dates on which they were checked.

| Site | Nr. of traps | Dates checked |
|------|-----|------|
| Kralický Sněžník[a] | 10 | 19.7., 23.7., 1.8., 9.8., 16.8., 26.8., 2.9. 2005 |
| Hrubý Jeseník - Šerák[a] | 7 | 19.7., 23.7., 1.8., 9.8., 16.8., 26.8., 2.9. 2005 |
| Hrubý Jeseník - Keprník[a] | 10 | 19.7., 23.7., 1.8., 9.8., 16.8., 26.8., 2.9. 2005 |
| Hrubý Jeseník - Červená hora[a] | 10 | 19.7., 23.7., 1.8., 9.8., 16.8., 26.8., 2.9. 2005 |
| Hrubý Jeseník – Malý Děd[b] | 7 | 20.7., 24.7., 2.8., 10.8., 17.8., 27.8., 3.9. 2005 |
| Hrubý Jeseník - Praděd[b] | 13 | 20.7., 24.7., 2.8., 10.8., 17.8., 27.8., 3.9. 2005 |
| Hrubý Jeseník - Vřesník[b] | 10 | 20.7., 23.7., 2.8., 9.8., 16.8., 26.8., 2.9. 2005 |
| Hrubý Jeseník - Vysoká hole[b] | 17 | 20.7., 24.7., 2.8., 10.8., 17.8., 27.8., 3.9. 2005 |
| **Total** | **84** | **7** |

**Notes.**
[a]Traps set on July 15, 2005.
[b]Traps set on July 16, 2005.

in 2005, and checked every week (Table 1). We also measured the distance of each trap from the tree-line (in m) using software ArcGIS 9 (*ESRI, 2004*) and the tree-line specification of *Treml & Banaš* (*2000*).

The yellow plastic traps provide a simpler and quicker way of acquiring large quantitative data sets than monitoring transects or capture-recapture. Using yellow water traps avoids human errors as the same principle determines the catch at all sites, which are all caught at the same time. This method is also very effective for faunal surveys of alpine habitats (including butterflies) as very few plants are in flower above the tree-line during the vegetation season (*Beneš, Kuras & Konvička*, *2000*).

This method is non-selective as different taxa of nectar feeding insects were caught. The contents of the traps were separated into different taxa, with the species and sex of the butterflies recorded (Table 2) and other insect groups conserved for future use. During the trapping period, only 15 species were caught: *Aglais urticae, Boloria dia, Coenonympha pamphilus, Erebia epiphron silesiana, Erebia euryale, Erebia ligea, Erebia sudetica sudetica, Gonepteryx rhamni, Inachis io, Issoria lathonia, Lycaena hippothoe, Pieris brassicae, Pieris napi, Pieris rapae* and *Polygonia c-album*. The species richness recorded is rather low, consisting mostly of species that migrate into alpine zones from lower altitudes. A total of 7,651 individuals were caught during the summer of 2005. *Erebia* spp. were the most abundant (94.2% of the individuals sampled), mainly because of their exclusive association with alpine and montane habitats and low probability of occurring at lower altitudes than the other species captured, which are generalists and good migrants. Therefore, we used only four species of *Erebia* (i.e., *E. epiphron silesiana, E. euryale, E. ligea, E. s. sudetica*) in the analysis and testing of the effects of environmental factors on autochthonous alpine butterfly assemblages.

## Classification of the vegetation

Vegetation at the alpine sites in the Hrubý Jeseník Mts. was classified according to the Habitat Catalogue of the Czech Republic (*Chytrý et al.*, *2010*), where habitats are specified in terms of the dominant, diagnostic and other plant species present. There are five types
**Table 2** Number of butterflies caught by the Moericke water traps in the High Sudetes Mts. during the summer of 2005.

| Alpine site/ Species | Kralický Sněžník | Šerák | Keprník | Červená hora | Malý Děd | Praděd | Mravenečník-Vřesník | Vysoká hole |
|---|---|---|---|---|---|---|---|---|
| *Aglais urticae* ♂♂ | 35 | 2 | 40 | 20 | 3 | 26 | 18 | 25 |
| *Aglais urticae* ♀♀ | 32 | 0 | 53 | 26 | 9 | 37 | 22 | 39 |
| *Boloria dia* ♀♀ | 0 | 0 | 0 | 0 | 1 | 0 | 0 | 0 |
| *Coenonympha pamphilus* ♀♀ | 0 | 1 | 0 | 0 | 0 | 0 | 0 | 0 |
| *Erebia epiphron silesiana* ♂♂ | 0 | 1 | 0 | 0 | 195 | 156 | 257 | 509 |
| *Erebia epiphron silesiana* ♀♀ | 0 | 0 | 0 | 0 | 96 | 70 | 76 | 149 |
| *Erebia euryale* ♂♂ | 893 | 264 | 76 | 591 | 379 | 534 | 594 | 287 |
| *Erebia euryale* ♀♀ | 405 | 109 | 20 | 391 | 222 | 307 | 289 | 182 |
| *Erebia ligea* ♂♂ | 0 | 1 | 0 | 14 | 1 | 0 | 27 | 0 |
| *Erebia ligea* ♀♀ | 0 | 0 | 0 | 7 | 1 | 1 | 26 | 0 |
| *Erebia sudetica sudetica* ♂♂ | 0 | 5 | 0 | 0 | 24 | 9 | 1 | 0 |
| *Erebia sudetica sudetica* ♀♀ | 0 | 0 | 0 | 0 | 30 | 12 | 0 | 0 |
| *Gonepteryx rhamni* ♂♂ | 0 | 0 | 0 | 0 | 0 | 1 | 0 | 1 |
| *Gonepteryx rhamni* ♀♀ | 0 | 0 | 0 | 1 | 0 | 0 | 0 | 0 |
| *Inachis io* ♂♂ | 0 | 0 | 1 | 0 | 1 | 0 | 0 | 1 |
| *Inachis io* ♀♀ | 0 | 0 | 0 | 1 | 0 | 0 | 0 | 0 |
| *Issoria lathonia* ♀♀ | 0 | 0 | 1 | 1 | 0 | 0 | 0 | 0 |
| *Lycaena hippothoe* ♀♀ | 0 | 0 | 0 | 0 | 0 | 0 | 0 | 1 |
| *Pieris brassicae* ♂♂ | 1 | 0 | 0 | 0 | 0 | 0 | 0 | 0 |
| *Pieris brassicae* ♀♀ | 0 | 0 | 0 | 0 | 0 | 0 | 1 | 1 |
| *Pieris napi* ♂♂ | 0 | 0 | 1 | 1 | 0 | 0 | 0 | 0 |
| *Pieris napi* ♀♀ | 0 | 0 | 0 | 1 | 1 | 0 | 0 | 1 |
| *Pieris rapae* ♂♂ | 0 | 2 | 1 | 0 | 2 | 1 | 2 | 3 |
| *Pieris rapae* ♀♀ | 1 | 3 | 4 | 2 | 3 | 1 | 2 | 3 |
| *Polygonia c-album* ♂♂ | 0 | 0 | 1 | 0 | 0 | 0 | 0 | 0 |
| *Polygonia c-album* ♀♀ | 0 | 0 | 1 | 0 | 0 | 1 | 0 | 0 |

of habitat at the alpine sites studied: (1) Alpine heathland dominated by *Calluna vulgaris*; (2) Subalpine *Vaccinium* vegetation dominated by *Vaccinium myrtillus* and *Vaccinium vitis-idaea*; (3) Alpine grassland dominated by *Avenella flexuosa*, *Nardus stricta* and *Festuca supina*; (4) Subalpine tall-herbaceous vegetation dominated by *Molinia coerullea*; and (5) *Pinus mugo* scrub dominated by dwarf pine. These habitats occurred at all the sites studied. We recorded the percentage vegetation cover made up of each type of habitat within an area of 10 m radius around each trap. The percentage cover of *Pinus mugo* was obtained from aerial photographs taken in 2005 with a grid of 0.3 m pixel resolution (*Wild, 2005*) (Table 3).

## DATA ANALYSIS

### Local effect of *Pinus mugo*

We used a unimodal direct gradient analysis, i.e., a CCA (Canonical Correspondence Analysis) to test the association of the abundances of the different species of *Erebia*

**Table 3  Altitude, area, extent of cover and origin of *Pinus mugo* in the alpine sites studied.**

| Alpine site | Altitude (m a.s.l.) | Area (ha) | *Pinus mugo* (%) | *P. mugo* origin[a] |
|---|---|---|---|---|
| Šerák | 1,351 | 21.9 | 36.0 | 0 |
| Mravenečník-Vřesník | 1,342 | 46.7 | 0 | 0 |
| Malý Děd | 1,368 | 55.0 | 24.7 | 0 |
| Červená hora | 1,333 | 65.7 | 35.7 | 0 |
| Keprník | 1,423 | 80.1 | 37.3 | 0 |
| Kralický Sněžník | 1,424 | 89.6 | 12.5 | 0 |
| Praděd | 1,492 | 142.5 | 22.1 | 0 |
| Vysoká hole | 1,464 | 678.5 | 14.3 | 0 |
| Krkonoše-East | 1,602 | 3212.6 | 46.0 | 1 |
| Krkonoše-West | 1,508 | 2116.1 | 57.0 | 1 |
| Harz | 1,142 | 131.0 | 5.0 | 0 |
| Babia Gora | 1,725 | 194.0 | 75.0 | 1 |

**Notes.**

[a] *Pinus mugo* origin: occurs naturally in the area = 1, planted or absent = 0.

with the environmental variables measured (*Ter Braak & Šmilauer*, *1998*). Vegetation types (*Calluna*, *Vaccinium*, *Avenella*, *Molinia* and *Pinus*), area of the alpine sites (ha) and distance to the tree-line (m) were considered as explanatory variables. There was little difference in the altitudes of the sites and as it had an insignificant effect it was not included in the model. The covariates of the model were selected by forward selection and were time of year (i.e., number of days from 1st July) and degree of isolation of each alpine site (i.e., distance to the nearest alpine site, in km). We configured the CCA model with an axis scaled in terms of inter-species distances. Species abundances were *log* transformed and results for males and females were identified in the analysis. Significance of canonical axis and explanatory variables was tested using a Monte-Carlo permutation test (5,000 restricted permutations). The permutation test was restricted by the split-plot design (time series and linear transect as a split-plot and freely exchangeable whole plot). Relationship between species abundance and cover of *Pinus mugo* was tested using a Generalized Linear Model (GLM) and a Poisson distribution. We tested both sexes of butterflies for possible differences in their behaviour (occupancy and usage of the habitat), especially for *Erebia* spp. (*Kuras, Beneš & Konvička*, *2001a*; *Konvička, Beneš & Kuras*, *2002*).

## Regional effect of *Pinus mugo*

Faunal structure and its historical development in the presence and absence of *Pinus mugo* was tested using methods based on diagnostic species of both butterflies and moths. Diagnostic species are alpine and boreal species with an exclusive association with alpine habitats or generally unlikely to be found below the tree-line (*Patočka & Kulfan*, *2009*). From an eco-zoogeographical point of view (*sensu Krampl*, *1992*) it means species with euboreal, boreo-alpine, arctic-alpine and subalpine distributions with disjunct and very narrow dispersal ranges within Central European mountains. All literature sources related to these mountain ranges were critically reviewed with regard to recent findings (*Soffner*, *1960*; *Krampl*, *1992*; *Jahn, Kozlowski & Pulina*, *1997*; *Liška*, *1997*; *Liška & Skyva*,

*1997*; *Liška*, *2000*; *Kuras et al.*, *2009*). In these publications, the following species were consistently claimed to be exclusively associated with the alpine environment and are therefore referred to as "diagnostic" in this paper (Table 4): *Argyresthia amiantella, Blastesthia mughiana, Callisto coffeella, Catoptria maculalis, Catoptria petrificella, Chioniodes viduellus, Clepsis rogana, Clepsis steineriana, Elachista dimicatella, Elachista kilmunella, Elophos operarius, Epichnopterix ardua, Erebia epiphron, Erebia sudetica, Glacies alpinatus, Incurvaria vetulella, Kessleria zimmermanni, Lampronia rupella, Olethreutes obsoletanus, Psodos quadrifarius, Rhigognostis senilella, Sparganothis rubicundana* and *Xestia alpicola*. *Pinus mugo* cover, nearest distances and areas of the alpine sites were estimated using aerial photographs and GIS analysis.

We used CCA to determine the effect of area of the alpine site, *Pinus mugo* cover and its origin (naturally occurring/planted) for all summits in the High Sudetes Mts., plus two geographically nearest and isolated alpine sites, i.e., Babia Gora Mts. (Slovakia) and Harz Mts. (Germany). Significance of the canonical axis and explanatory variables was tested using a Monte-Carlo permutation test (5,000 unrestricted permutations).

Next, we tested the similarity in the species composition at all the sites. Diagnostic species were allocated values 1/0 (present/absent) and tested using the UPGMA (unweighted pair group using arithmetical average) clustering method (*Everitt & Hothorn*, *2006*). Significance of each cluster was estimated and possible subjectivity of the diagnostic species list decreased using bootstrap methods, so that a dissimilarity matrix of the alpine sites was generated for each cluster and bootstrap probabilities calculated using multi-scale bootstrap re-sampling, in order to acquire AU (Approximately Unbiased) *p*-values.

## RESULTS

### Local effect of *Pinus mugo*

After filtering out the effects of covariates, the CCA model explained 15.0% (adjusted explained variation 13.5%) of the variance of the results for this species. As shown in the ordination graph (Table 5 and Fig. 3), most of the variability is accounted for in terms of the distance to the tree-line, area of the alpine site and vegetation cover of *Avenella* and *Pinus*. With increase in the area of the site we recorded an increase in the abundance of *Erebia epiphron*, a typical alpine species. Other *Erebia* spp. are negatively correlated with the area of the site.

Using the CCA model, we also tested each parameter separately. Each environmental variable was included in the model individually and remaining parameters acted as covariates. We recorded that individual environmental parameters (after subtracting summarized variability of the covariates) accounted for the variability as follows: time of year 11.9% , distance from the tree-line 7.1%, distance to the next nearest alpine site 2.7%, area of alpine site 3.1%, vegetation cover of *Calluna* 0.5%, *Avenella* 0.9%, *Pinus* 0.9%, *Molinia* 0.4% and *Vaccinium* 0.4%.

The association between species abundance and area covered by *Pinus mugo* was further tested using a Generalized Linear Model. The mountain specialists (*Erebia* spp.) are affected more than the generalists (*Aglais urticae* and *Pieris rapae*) by *Pinus mugo* (Table 6). The

**Table 4** Diagnostic species in the High Sudetes Mts.: the Kralický Sněžník Mt., the Hrubý Jeseník Mts. and the Krkonoše Mts. (Czech Republic), and the two geographically nearest alpine habitats: Harz (Germany) and Babia Gora (Slovakia/Poland).

| Alpine site/ Species | Kralický Sněžník | Šerák | Keprník | Červená hora | Malý Děd | Mravenečník Vřesník | Praděd | Vysoká hole | Krkonoše-West | Krkonoše-East | Harz | Babia Gora |
|---|---|---|---|---|---|---|---|---|---|---|---|---|
| *Incurvaria vetulella* | + | | | | | | + | + | + | + | | |
| *Lampronia rupella* | | | | | | | + | + | + | + | | + |
| *Argyrhestia amianthella* | | | | | | | | | + | + | | |
| *Rhigognostis senilella* | + | | | | | | | | + | + | | + |
| *Elachista kilmunella* | + | | | | | | | | + | + | + | |
| *Elachista dimicatella* | | | | | | | | | | | | + |
| *Chioniodes viduellus* | + | | | | | | + | | + | + | | |
| *Sparganothis rubicundana* | + | | + | | | | + | + | | | | |
| *Clepsis steineriana* | | | | | + | | + | + | | | | |
| *Clepsis rogana* | + | + | + | + | + | + | + | + | | | + | |
| *Blastesthia mughiana* | | | | | | | | | + | + | | |
| *Olethreutes obsoletanus* | | | | | | | | | | + | | |
| *Catoptria maculalis* | | | | | | | | | + | + | | |
| *Catoptria petrificella* | | | + | + | + | + | + | + | | | | |
| *Erebia epiphron*[a] | | | | | + | + | + | + | | | + | |
| *Erebia sudetica* | + | + | | + | + | + | + | + | | | | |
| *Psodos quadrifarius* | | | | | | | | | + | + | | |
| *Glacies alpinatus* | | | | | + | | + | + | + | + | | |
| *Epichnopterix ardua*[b] | + | | | | + | | + | + | + | + | + | |
| *Xestia alpicolla* | | | | | | | + | | | + | | |
| *Elophos operarius* | | | | | | | | | | + | | + |
| *Callisto coffeella* | | | | | | | | | | + | | |
| *Kessleria zimmermanni* | | | | | | | | | | + | | |

**Notes.**
[a] *Erebia epiphron* was introduced into the Krkonoše Mts. in 1932 (*Kuras et al.*, *2001b*; *Schmitt, Čížek, & Konvička*, *2005*), thus this species is not included in the analysis of the results for the Krkonoše Mts.
[b] Several authors mention this species as *E. sieboldi* (cf. *Laštůvka & Liška*, *2011*).

**Table 5  Summary of Monte Carlo test of the CCA model "species ∼ env. variables: area of alpine site (ha), distance to the tree-line (m), cover of the different types of vegetation; covariates: time of year, distance to the nearest alpine site (km)" used to determine the local effect.**

| Axes | I. | II. | III. | IV. |
|---|---|---|---|---|
| Eigenvalues | 0.117 | 0.039 | 0.011 | 0.002 |
| Species-environment correlations | 0.620 | 0.439 | 0.222 | 0.139 |
| Cumulative percentage variance | | | | |
| of species data | 10.3 | 13.7 | 14.7 | 14.9 |
| of species-environment relation | 68.9 | 91.7 | 98.0 | 99.4 |
| Sum of all eigenvalues | | 1.134 | | |
| Sum of all canonical eigenvalues | | 0.150 | | |
| Test of significance of first canonical axis | $F$-ratio = 47.419 | | $P$-value = 0.0012 | |
| Test of significance of all canonical axes | $F$-ratio = 10.376 | | $P$-value = 0.0012 | |

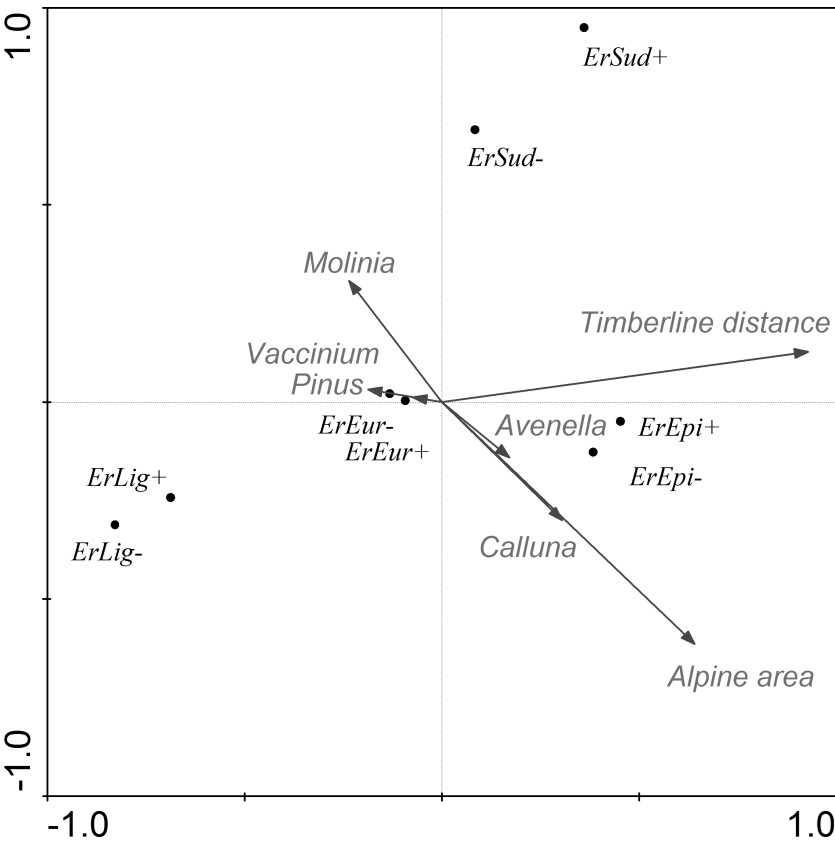

**Figure 3  CCA ordination graph with the environmental variables and butterflies that occurred at the alpine sites in the Kralický Sněžník and Hrubý Jeseník Mts.** Length of the arrows indicate the amount of variance accounted for by a particular variable. Species positions are indicated by black dots in the reduced ordination area of the first and second canonical axes. Acronyms for the species: *ErEpi, Erebia epiphron silesiana; ErEur, Erebia euryale; ErLig, Erebia ligea; ErSud, Erebia sudetica sudetica;* +, female; −, male.

**Table 6 Results of GLM regressions of the abundances of the different butterflies and percentage cover of *Pinus mugo*.**

| Species | Beta | SE | *T*-value | F | p | AIC |
|---|---|---|---|---|---|---|
| *Aglais urticae* ♂♂ | −0.014 | 0.004 | −3.75 | 3.85 | >0.05 | 1077.618 |
| *Aglais urticae* ♀♀ | −0.012 | 0.003 | −3.81 | 3.96 | <0.05 | 1297.152 |
| *Erebia ligea* ♂♂ | −0.005 | 0.006 | −0.88 | 0.17 | >0.05 | 530.437 |
| *Erebia ligea* ♀♀ | −0.062 | 0.020 | −3.16 | 7.27 | <0.01 | 389.823 |
| *Er. epiph. silesiana* ♂♂ | −0.027 | 0.002 | −15.07 | 12.45 | <0.01 | 7035.796 |
| *Er. epiph. silesiana* ♀♀ | −0.019 | 0.003 | −7.35 | 7.94 | <0.01 | 2445.229 |
| *Erebia euryale* ♂♂ | −0.010 | 0.001 | −15.70 | 6.17 | <0.05 | 1.51e+004 |
| *Erebia euryale* ♀♀ | −0.015 | 0.001 | −15.26 | 18.68 | <0.01 | 5853.988 |
| *Er. sudet. sudetica* ♂♂ | 0.000 | 0.005 | 0.09 | 0.00 | <0.05 | 620.972 |
| *Er. sudet. sudetica* ♀♀ | −0.119 | 0.034 | −3.54 | 8.47 | <0.01 | 614.618 |
| *Pieris rapae* ♂♂ | 0.020 | 0.008 | 2.67 | 3.53 | >0.05 | 165.006 |
| *Pieris rapae* ♀♀ | 0.003 | 0.009 | 0.34 | 0.05 | >0.05 | 217.394 |

**Notes.**
Key: Model resid. DF (all models) = 442, Link function Log, Poisson's distribution. "Beta" indicates the slope of regression line.

**Table 7 Summary of Monte Carlo test of CCA model "species ∼ env. variables: area of alpine site, % cover of *Pinus mugo* and its origin" for determining the regional effect.**

| Axes | I. | II. | III. | IV. |
|---|---|---|---|---|
| Eigenvalues | 0.610 | 0.398 | 0.079 | 0.224 |
| Species-environment correlations | 0.950 | 0.883 | 0.784 | 0.000 |
| Cumulative percentage variance | | | | |
| of species data | 31.3 | 51.8 | 55.9 | 67.4 |
| of species-environment relation | 56.1 | 92.7 | 100.0 | 0.0 |
| Sum of all eigenvalues | | 1.947 | | |
| Sum of all canonical eigenvalues | | 1.087 | | |
| Test of significance of first canonical axis | *F*-ratio = 3.646 | | *P*-value = 0.0120 | |
| Test of significance of all canonical axes | *F*-ratio = 3.374 | | *P*-value = 0.0024 | |

abundance of the two most abundant specialists (*Erebia epiphron* and *Erebia euryale*) is considerably reduced in areas where there is a relatively low cover of *Pinus mugo* (Fig. 4). We also tested the association of both sexes of these butterflies with the area covered by *Pinus mugo,* which revealed no significant difference between males and females in spite of the higher numbers of males caught.

## Regional effect of *Pinus mugo*

CCA accounted for 61% of the variance in the species' environment (i.e., area of alpine site, percentage cover and origin of *Pinus mugo*) (Table 7, Fig. 5). The effect of *Pinus mugo* origin and area of alpine site were significant, but not the percentage cover of *Pinus mugo*.

Relationship between species and environmental variables identified by CCA was further tested using a similarity analysis (Fig. 6) and three different groups of alpine sites

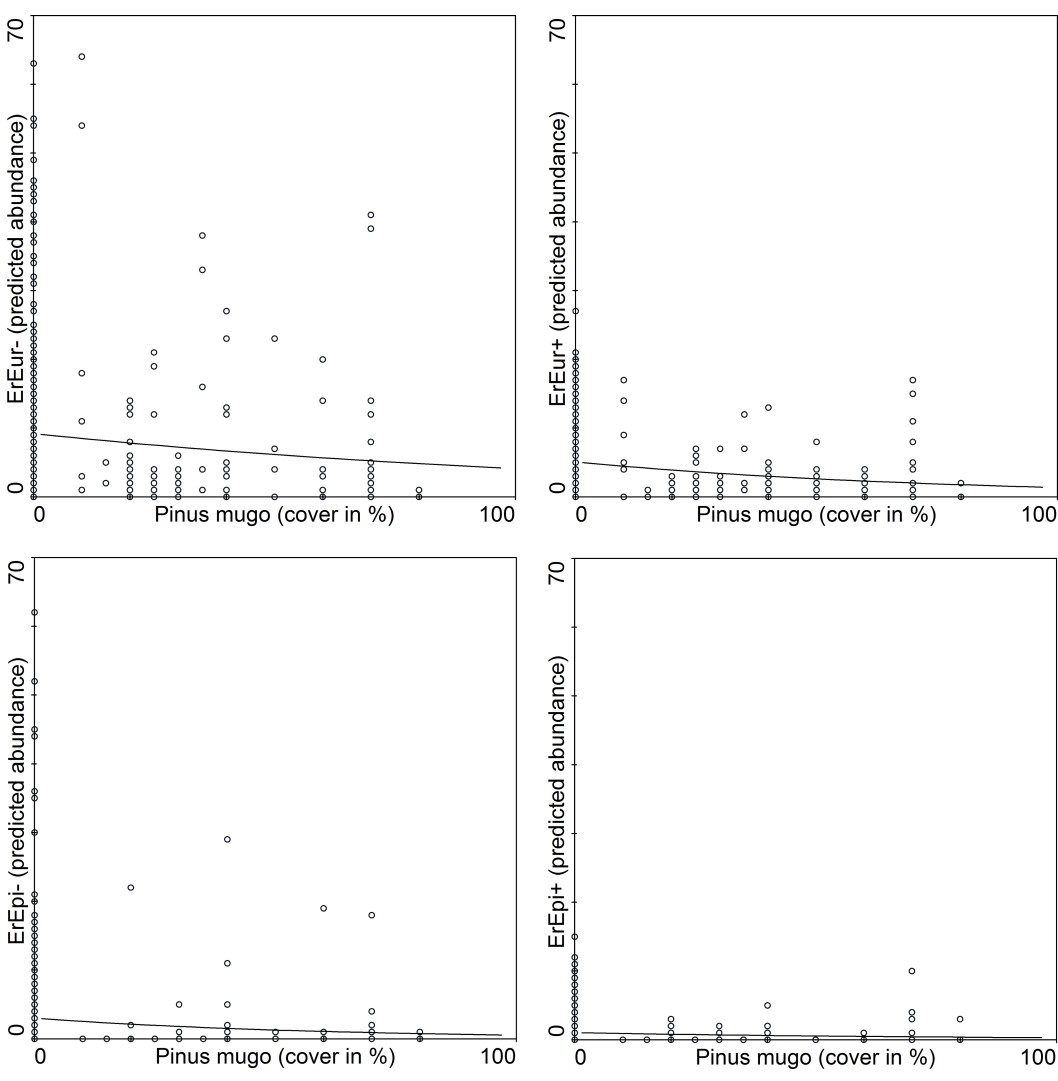

**Figure 4** **GLM regressions of the relationships between the two most abundant species of butterfly and *Pinus mugo* cover [%].** Species names are abbreviated and differentiated according to gender: *ErEpi, Erebia epiphron silesiana; ErEur, Erebia euryale*; +, female; −, male.

were identified. The first cluster is of alpine habitats with naturally occurring *Pinus mugo* (Krkonoše-West and Krkonoše-East), which are clearly distinguished from other sites (Multi-scale Bootstrap Re-sampling, AU = 98, p = 0.01). The second cluster includes alpine summits in the Hrubý Jeseník Mts., i.e., Vysoká hole, Praděd and Malý Děd, which is presumably associated with the presence of planted *Pinus mugo* and their large areas. Furthermore, we can also consider the cluster that includes Kralický Sněžník, Harz, Keprník, Červená hora, Šerák and Mravenečník-Vřesník (last four are in the Hrubý Jeseník Mts.), which are small areas with a dense cover of planted *Pinus mugo*. These two clusters were separated at a level of significance AU = 92, p = 0.07. The position of the Babia Gora Mt. in the dendrogram seems to be ambiguous (AU = 86, p = 0.11) due to a combination of small species pool, small alpine area and the presence of naturally occurring *Pinus mugo*.

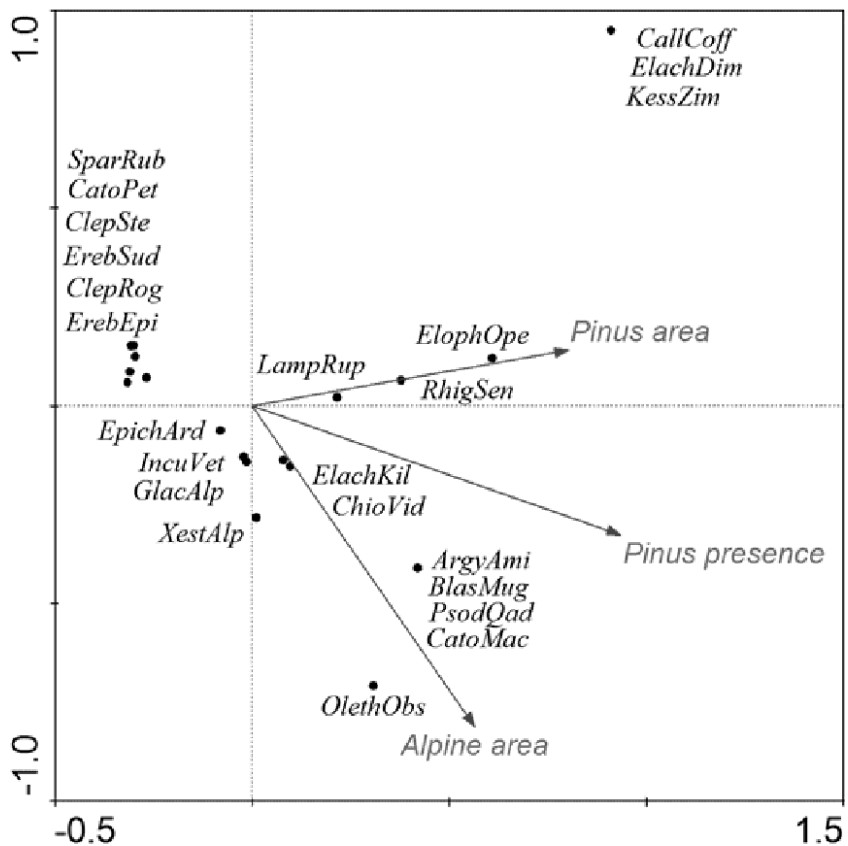

**Figure 5** CCA ordination graph, with the environmental variables and Lepidopteran species that occurred at the alpine sites in the High Sudetes Mts. (CZ), Babia Gora Mts. (SK) and Harz Mts. (D). Species positions are indicated by black dots in the reduced ordination area of the first and second canonical axes. Acronyms for species: *ArgyAmi, Argyresthia amiantella; BlastMug, Blastesthia mughiana; CallCof, Callisto coffeella; CatoMac, Catoptria maculalis; CatoPet, Catoptria petrificella; ChioVid, Chioniodes viduellus; ClepRog, Clepsis rogana; ClepSte, Clepsis steineriana; ElacDim, Elachista dimicatella; ElacKil, Elachista kilmunella; ElopOpe, Elophos operarius; EpicArd, Epichnopterix ardua; ErebEpi, Erebia epiphron; ErebSud, Erebia sudetica; GlacAlp, Glacies alpinatus; IncuVet, Incurvaria vetulella; KessZim, Kessleria zimmermanni; LampRup, Lampronia rupella; OletAbs, Olethreutes obsoletanus; PsodQua, Psodos quadrifarius; RhigSen, Rhigognostis senilella; SparRub, Sparganothis rubicundana; XestAlp, Xestia alpicola.*

## DISCUSSION

### Local effect of *Pinus mugo*

In a local context, we tested for the recent effect of *Pinus mugo* on the butterfly assemblage, namely it's overgrowing of alpine sites and competing with native vegetation. We tested the response of males and females separately to the vegetation cover because previous research showed that males fly more frequently in search of food than females, which mainly bask and feed on nectar (*Kuras, Beneš & Konvička, 2001a*; *Konvička, Beneš & Kuras, 2002*). However, our analyses did not indicate any significant differences between the sexes of *Erebia spp.* except for the greater probability of catching males.

The presence of *Erebia spp* was most significantly associated with the area of an alpine site and its distance to the tree-line, followed by vegetation dominated by *Avenella* (a characteristic species of alpine tundra) and *Pinus mugo*. Correlation of *E. epiphron*

## Cluster dendrogram with AU/BP values (%)

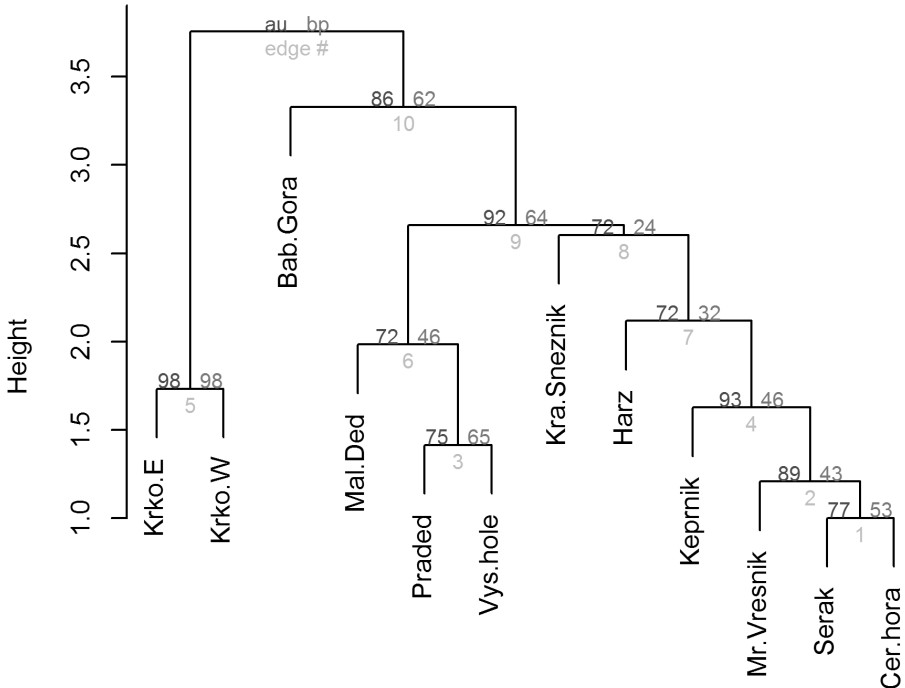

Distance: euclidean
Cluster method: average

**Figure 6** **Cluster analysis (dendrogram) indicating the differences between alpine sites in the Central European middle high mountains calculated on the basis of the occurrence of diagnostic species of Lepidoptera (see chapter Methods–Data Analysis) and a clustering method UPGMA (unweighted pair group using arithmetical average).** AU/BP values: AU (Approximately Unbiased) *p*-value is more accurate than the BP value as an unbiased *p*-value and is computed by multiscale bootstrap resampling; BP (Bootstrap Probability) value is a simple statistic computed by bootstrap resampling and tends to have a biased *p*-value when the absolute value of c (i.e., the value related to geometric aspects of hypotheses) is large. Site codes: Krko.E, Krkonoše-East; Krko.W, Krkonoše-West; Bab.Gora, Babia Gora; Mal.Ded, Malý Děd; Praded, Praděd; Vys.hole, Vysoká hole; Kra.Sneznik, Kralický Sněžník; Harz, Harz; Keprnik, Keprník; Mr.Vresnik, Mravenečník-Vřesník; Serak, Šerák; Cer.hora, Červená hora).

abundance with the area of the sites supports an alpine origin for this species as does its association with alpine grassland where vegetation dominated by *Avenella* and *Calluna* prevail. At small sites there are fewer *E. epiphron* and *E. sudetica* (*Kuras et al., 2003*), species with a high association with treeless sites, or these species do not occur there (*Kuras et al., 2001b*). Thus, the area of an alpine site alone seems to be of greater importance for the butterfly assemblage than its heterogenity in terms of different types of alpine vegetation. The congeneric species, *E. Euryale* and *E. Ligea,* also occur at alpine sites, but mainly at lower altitudes, i.e., in mountain taiga (*Kuras, Beneš & Konvička, 2000*), or like *E.s. sudetica* near springs close to the upper tree-line (*Kuras et al., 2001b*). This fact accounts for the significant effect of distance to the tree-line on the species studied. The relatively small

effect of site heterogenity can be due to the similarity of the different types of vegetation and high mobility of the butterflies (*Chytrý et al.*, *2010*).

Even though adult butterflies are highly mobile *Pinus mugo* serves as a barrier to their dispersal (*Kuras, Beneš & Konvička*, *2001a*; *Konvička, Beneš & Kuras*, *2002*). This is also supported by Fig. 4. Dwarf pine overgrows alpine vegetation and in this way is changing the characteristics of this habitat (food available for adults and quantity of suitable plants for caterpillars—see *Zeidler et al.*, *2012*) and decreasing the area of native grass-herbaceous vegetation at alpine sites. Over the last 30 years the extent of the cover of planted dwarf pine has increased by 63% and the percentage cover at small alpine sites in the Hrubý Jeseník Mts. is currently more than 30% and is continuing to increase (*Treml et al.*, *2010*).

## Regional effect of *Pinus mugo*

*Pinus mugo* significantly affected the fauna and flora of alpine sites in the Central European Mountains during their postglacial development (*Jeník*, *1961*; *Gutierrez*, *1997*; *Čížek et al.*, *2003*). We found a significant effect of the presence of *Pinus mugo*, but not the area covered by this species at alpine sites. We suppose this is a consequence of the postglacial development at the sites studied. The tree-line shifted during the postglacial era and the area of treeless sites obviously changed (*Treml & Banaš*, *2008*). During the Holocene climate optimum, the conditions for alpine species were even less favourable than now because the mountain summits were overed with dwarf pine or spruce (*Rybníček & Rybníčková*, *2004*). Therefore, species confined to treeless habitats could survive only at very few localities, e.g., glacial cirques, stone debris fields, rocks or wind swept summits.

We also recorded significant differences in the Lepidopteran fauna at the alpine sites studied in the High Sudetes Mts., which were associated with differences in the historical occurrence of *Pinus mugo*. Our results (Fig. 6) are clearly similar to the effect that *Pinus mugo* had on the fauna of alpine sites during the Holocene. Alpine sites with naturally occurring *Pinus mugo* (Krkonoše Mts, Babia Gora Mt.), which most probably completely covered or at least a large part of the summits of these mountains in the past, host predominantly species of Lepidoptera that are able to feed on *Pinus* or those that survived in glacial cirques or stone debris fields. Alpine sites where dwarf pine was not present (all summits of the Hrubý Jeseník Mts., Kralický Sněžník Mt. and Harz Mt.) were covered with dwarf spruce, which does not form such dense formations as dwarf pine. Thus, species of Lepidoptera associated with grassy tundra could survive there until colder conditions resulted in the tree line occurring at lower altitude.

These changes selected the species assemblages currently recorded at the different alpine sites and the most critical time was when the summits were either completely or nearly covered by trees during the Holocene. Recent climate changes favoured pine, which became a more prominent element in the landscape again and so reversed a natural multicentennial or even millennial trend in tree line decline and recession (*Kullman*, *2007*). Global warming is now favouring a similar spread of planted *Pinus mugo* into alpine sites where it previously did not occur. Even though its current coverage at these sites is not very extensive, *Pinus mugo* has the potential to change these treeless sites and increase their degree of isolation (*Bílá et al.*, *2012*).

## CONCLUSIONS

The spread of *Pinus mugo* into alpine treeless sites is rapid as it covers approximately another 2% of the alpine area in the High Sudetes each year (*Zeidler et al.*, *2012*). Dwarf pine together with spruce spread into grassy tundra and out compete the native fauna and flora. In addition, dwarf pine can also colonize glacial cirques. Because of its competitive ability and fast growth, *Pinus mugo* (*Wild & Wildova*, *2002*; *Wild*, *2005*) is likely to rapidly colonize the alpine zone, particularly sites in areas where *Pinus mugo* was planted. The most threatened are small alpine sites in the Hrubý Jeseník Mts., where the authorities responsible for this Protected Landscape Area have already started to remove dwarf pine. This study also helped to identify sites with the highest priority and the process of eradicating dwarf pine at these sites is ongoing. Other effects of climate change on Lepidoptera, such as changes in species and sources of food, or population dynamics, would be interesting topics for further research.

## ACKNOWLEDGEMENTS

The authors thank Václav Treml for providing the data on position of the tree-line and locality of alpine sites in the High Sudetes Mts., and for his permission to print maps of the study area (Figs. 1 and 2). We also thank Tony Dixon for improving the English of this paper.

### Funding

This research was funded by the Ministry of Environment of the Czech Republic (project VaV SM/6/70/05), grant No. IG UP 913104041/31 and by the MŠMT grants LC06073 and LO1415. The funders had no role in study design, data collection and analysis, decision to publish, or preparation of the manuscript.

### Grant Disclosures

The following grant information was disclosed by the authors:
Ministry of Environment of the Czech Republic: IG UP 913104041/31.
MŠMT: LC06073, LO1415.

### Competing Interests

The authors declare there are no competing interests.

### Author Contributions

- Karolína Bílá conceived and designed the experiments, performed the experiments, analyzed the data, contributed reagents/materials/analysis tools, wrote the paper, prepared figures and/or tables, reviewed drafts of the paper.
- Jan Šipoš analyzed the data, contributed reagents/materials/analysis tools, prepared figures and/or tables, reviewed drafts of the paper.
- Pavel Kindlmann wrote the paper, reviewed drafts of the paper.

- Tomáš Kuras conceived and designed the experiments, analyzed the data, contributed reagents/materials/analysis tools, wrote the paper, prepared figures and/or tables, reviewed drafts of the paper.

**Field Study Permissions**

The following information was supplied relating to field study approvals (i.e., approving body and any reference numbers):

This field survey was approved by the Czech Ministry of Environment and we were allowed to collect Lepidoptera species on all studied alpine sites which are part of the Protected Landscape area Jeseníky. The permission was issued for a group of scientists, one of which was Dr. Tomáš Kuras. Permission Nr.: MŽP/13341/04-620/2319/04 from the 21st of July 2004, valid until the 30st of November 2007.

**Data Availability**

The research in this article did not generate any raw data.

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
