# Peer review of "Consequences for selected high-elevation butterflies and moths from the spread of Pinus mugo into the alpine zone in the High Sudetes Mountains"

_PeerJ, doi:10.7717/peerj.2094_

## Round 0.1 · original submission · Major Revisions

As you can see, the reviewers are both favourable, but they both raise a number of points. My sense is that you can address these quickly and thus I encourage yo to do so. The standard 55 days (below) will put this well into my field season, when my ability to make a final decision will be limited. So, if you can do this quickly, that would help!

·

Basic reporting

This paper reports on butterfly communities associated with isolated relict alpine grassland habitats situated in mountain areas of moderate elevation in Central Europe. Climate change and forestry (viz. introduction of pine in earlier decades) have facilitated the expansion of dwarf pine there. The authors wish to establish whether this expanding invasive woody plant species might threaten local butterfly assemblages.

The paper is well written, the context of the study is well laid out and supported with references and maps. The overall structure of the paper is OK.

Figures are all relevant, but I suggest improving their quality. For example, avoid abbreviations for predictor vectors in the ordination diagrams. It would also be helpful for readers to see indicated by some symbol which of the vectors in the ordination diagrams carry more variance than expected by chance (according to the Monte Carlo permutation tests reported in the text).

In Fig. 4, the x-axis is not logical. How could pine cover become negative at all? This axis must start with zero % and end with 100 %! In this same figre, what is the unit for "response" on the y-axis? This should be very explicitly said. Finally, I always prefer to see the real data points being shown, and not only the regression lines.

Site codes in Fig. 6 (the cluster dendrogram) should strictly be identical to those used in Table 4 (or vice versa). Please take care that these codes are fully coherent throughout the paper!

Overall, this is a technically sound and well described study - which has been very substantially improved relative to an earlier version that, by chance, I had for review some months ago.

Experimental design

The design of the study is clear. At the spatial scale of the Czech mountain tops, only the Erebia ringlets have been evaluated - they accounted for the vast majority of records and are also ecologically bound to grassland habitats in mountain areas. Thus, it is useful to focus on them. On the regional scale, incidence data have been assembled for a wide range of Lepidopteran species characteristic for high elevation habitats of this type of mountains. Accordingly, the analysis has taken a different route here.

Research questions are well defined, and the research fills an identified knowledge gap. the use of yellow traps for butterfly community assessment may appear a bit unusual, but since the analysis is focused on Erebia species which obviously were recorded easily with that approach, there is nothing wrong with that.

Statistical methodology is adequate and state of the art.

Validity of the findings

I found the results convincingly worked out. Beyond the more narrow community of researchers interested in butterfly ecology, the findings are meaningful also in the context of conservation biology and climate change research. The data is robust, statistically sound, & controlled.

I was a bit disappointed to see that, while in Erebia ringlets the two sexes have been evaluated separately with regard to their response to pine cover and other site descriptors, this thread has not been taken up again in the discussion. Are there any cases where your data indicate that sexes respond differently? Fig 3 suggests this at best to be the case with E. sudetica. Why could that be so, what differences in habitat requirements between the sexes could be reflected here?

Otherwise, I found the conclusions and interpretations sound and not exceeding what has been demonstrated through the data and analyses.

·

Basic reporting

This work is fundamentally sound although confusing in parts. How significant the results are is debatable although given the hothouse nature of conservation debates in Europe it probably deserves to be published.
First the authors have shown clear effects of the spread of Pinus mugo on three alpine species of Erebia butterfly (only). All the other species they encountered are generalists with wide altitudinal and ecosystematic ranges, many of which are also noted for their long-distance movement capacity. That the spread of an overgrowing shrub is impacting upon sedentary species whose larvae feed on grasses is not surprising although the documentation is sound.

Experimental design

The work on butterflies is sound although using any form of trap for butterflies is notoriously variable and selective. Most butterfly surveys rely on transect walks rather than traps - but with this caveat the results are sound - especially (perhaps only) for Erebia butterflies which seem well sampled by the water traps.

The work on 'diagnostic' species of (mainly) moths (lines 217-222) is unexplained. Where did this distributional data come from - certainly not from any trapping effort of the authors. I presume it is a meta-analysis based on published atlas data somewhere - if the authors explain this then I have missed that explanation somewhere.

In a similar vein authors need be careful about claiming levels of generality for their results - butterflies are a small proportion of all Lepidoptera - as are the small group of 'diagnostics' they analyse further. These are not 'Lepidoptera communities' - whatever that means - but highly selected assemblages from within the Lepidoptera.

Validity of the findings

The findings, as far as they go, are valid. There is a temptation, not resisted by these authors, to regard ordinations as demonstrating relationships between multivariate data and putative driving variables. This is illusory. At best these ordinations allow hypotheses to be proposed for further testing - preferably by manipulation (which would be easily - if not legally - done in this case). There is nothing wrong with this very common approach using ordinations to rank driving variables (I have used it myself often) but the authors state clearly as to the limits of their analytical approach and the conclusions they draw from it.

The conservation consequences of their findings are not discussed. The spread of Pinus mugo seems a natural response to warming - what lepidopteran species does it bring with it (several species of Thera for a start) - will these enrich the communities - the no change/no loss approach to conservation especially in alpine zones is surely a policy of despair - an interesting discussion to be had!

Additional comments

Most of my comments for the authors are embedded above. The English needs a little editorial attention altho' meaning is generally clear.

---

## Round 0.2 · accepted · Accept

You have satisfied the reviewers in my estimation. But, the point of the title is well taken. Please change it!

·

Basic reporting

I have not much to add in comparison with my comments on the first version. The changed title is misleading, since the second part of the paper (the inter-mountain comparison across Central European regions) mostly is concerned with moths, NOT butterflies. Suggestion for improved title:
Consequences for selected high-elevation butterflies and moths of the spread of Pinus mugo into the alpine zone in the High Sudetes Mts.

Experimental design

This is well explained and now fits to the goals of the study, in its revised version.

Validity of the findings

The findings are important for the conservation biology of alpine habitats, above the tree line, in those lower mountain areas where only very limited amount of such habitats does exist as relict from the (post-)glacial periods. The results are based on solid statistical evaluation of sufficiently large and representative samples.

Additional comments

The revision was very well done. To my perception, all important issues raised by the two reviewers have adquately been dealt with.